

# Wearable technology to inform the prediction and diagnosis of cardiorespiratory events: a scoping review

Hamzeh Khundaqji[1], Wayne Hing[1], James Furness[1] and Mike Climstein[2,3]

[1] Faculty of Health Sciences & Medicine, Bond University, Gold Coast, Queensland, Australia
[2] Faculty of Health, Southern Cross University, Bilinga, Queensland, Australia
[3] Physical Activity, Lifestyle, Ageing and Wellbeing Faculty Research Group, University of Sydney, Sydney, New South Wales, Australia

Corresponding author
Hamzeh Khundaqji,
hamzeh.khundaqji@student.bond.edu.au

## ABSTRACT

**Background**. The need for health systems that allow for continuous monitoring and early adverse event detection in individuals outside of the acute care setting has been highlighted by the global rise in chronic cardiorespiratory diseases and the recent COVID-19 pandemic. Currently, it is unclear what type of evidence exists concerning the use of physiological data collected from commercially available wrist and textile wearables to assist in clinical decision making. The aim of this review was therefore to systematically map and summarize the scientific literature surrounding the use of these wearables in clinical decision making as well as identify knowledge gaps to inform further research.

**Methodology**. Six electronic bibliographic databases were systematically searched (Ovid MEDLINE, EMBASE, CINAHL, PubMed, Scopus, and SportsDiscus). Publications from database inception to May 6, 2020 were reviewed for inclusion. Non-indexed literature relevant to this review was also searched systematically. Results were then collated, summarized and reported.

**Results**. A total of 107 citations were retrieved and assessed for eligibility with 31 citations included in the final analysis. A review of the 31 papers revealed three major study designs which included (1) observational studies ($n = 19$), (2) case control series and reports ($n = 8$), and (3) reviews ($n = 2$). All papers examined the use of wearable monitoring devices for clinical decisions in the cardiovascular domain, with cardiac arrhythmias being the most studied. When compared to electrocardiogram (ECG) the performance of the wearables in facilitating clinical decisions varied depending upon the type of wearable, user's activity levels and setting in which they were employed. Observational studies collecting data in the inpatient and outpatient settings were equally represented. Eight case control series and reports were identified which reported on the use of wrist wearables in patients presenting to an emergency department or clinic to aid in the clinical diagnosis of a cardiovascular event. Two narrative reviews were identified which examined the impact of wearable devices in monitoring cardiovascular disease as well as potential challenges they may pose in the future.

**Conclusions**. To date, studies employing wearables to facilitate clinical decisions have largely focused upon the cardiovascular domain. Despite the ability of some wearables to collect physiological data accurately, there remains a need for a specialist physician

to retrospectively review the raw data to make a definitive diagnosis. Analysis of the results has also highlighted gaps in the literature such as the absence of studies employing wearables to facilitate clinical decisions in the respiratory domain. The disproportionate study of wearables in atrial fibrillation detection in comparison to other cardiac arrhythmias and conditions, as well as the lack of diversity in the sample populations used prevents the generalizability of results.

## INTRODUCTION

The transition towards patient-centric, personalized healthcare has prompted rapid advances in wearable devices and mobile cloud computing technologies. The need for systems that allow the continuous monitoring of individuals outside of the acute setting has been highlighted by the global rise in noncommunicable diseases (NCDs) and more so by the recent COVID-19 pandemic. The steady increase in NCDs, particularly those of cardiovascular and chronic respiratory nature, have emphasized the need to provide patients with home-based monitoring, clinical care, and support to off-load burden from primary care practices as well as support outpatient observational research (*Roemer et al., 2020*; *World health statistics, 2020*).

Advancements in the production of micro-electromechanical systems and conductive threads has led to the rise in wearable monitoring technologies such as wrist wearables and intelligent textiles. Decreasing costs paired with improved connectivity and reliability have also led to their widespread commercial adoption. Wrist technology such as the Fitbit or Apple Watch have traditionally been employed in non-clinical settings to monitor simple physiological metrics such as heart rate (HR) with an emphasis on general health and fitness (*Thomson et al., 2019*; *Bai et al., 2018*). With increasing validity and reliability of intelligent textiles they have begun to be adopted by the academic community as non-invasive monitoring tools for a plethora of physiological metrics in research (*Khundaqji et al., 2020b*). Moreover, developments in deep learning, a branch of machine learning, have demonstrated increasing promise for the clinical use of wearables in healthcare. The integration of wearable technology and deep learning algorithms into the clinical pathway may assist in the processing and analysis of immense volumes of data to potentially aid in novel disease phenotyping, disease surveillance, and complex decision making (*Waring, Lindvall & Umeton, 2020*). Deep learning algorithms have been successfully employed in clinical applications such as predicting the risk of lung cancer (*Ardila et al., 2019*), diagnosing pneumonia from chest X-rays (*Rajpurkar et al., 2018*; *Hashmi et al., 2020*) and identifying patients at high risk of being transferred to the intensive care unit (*Escobar et al., 2016*). However, currently within the realm of wearables, most data collected is not used to build predictive models that are successively integrated in the clinical setting.

As identified in previous work by the authors, due to the infancy of the field, the current body of knowledge surrounding wearables is mainly centered towards technical aspects such as design, reliability and validity in controlled settings (*Khundaqji et al., 2020b*). Although this type of evidence continues to be important, the next phase towards clinical adoption will be the ability to accurately and reliably transform the physiological data collected by wearables into a meaningful clinical decision.

Nevertheless, it is currently unclear what type of evidence exists concerning the use of physiological data collected from commercially available wearables in the form of wrist wearables and intelligent textiles in clinical decision making. For this purpose, a scoping review was undertaken to systematically survey the existing scientific literature on the use of wrist wearables and intelligent textiles for clinical decision making. The clinical area of focus was limited to cardiovascular and respiratory clinical decisions as these account for a large proportion of chronic conditions and as identified by previous research are among the most suitable to be monitored by existing wearable sensor technology (*Khundaqji et al., 2020b*).

The primary aims were to (1) provide a clear indication of the types and volume of the scientific literature concerning the use of wrist wearables or intelligent textiles in clinical decision making, (2) summarize the research completed to date, and (3) identify knowledge gaps to inform further research.

## SURVEY METHODOLOGY

### Protocol and registration

An a priori protocol was developed using the Preferred Reporting Items for Systematic Reviews and Meta-analysis Extension for Scoping Reviews: Checklist and Explanation (PRISMA-Scr) (*Tricco et al., 2018*). The final protocol was registered prospectively with the Open Science Framework (https://osf.io/37byq/) on September 20, 2020 (*Khundaqji et al., 2020a*).

The eligibility criteria were informed by the Population-Concept-Context framework recommended by the Joanna Briggs Institute (JBI) Reviewer's Manual (*Aromataris & Munn, 2020*).

### Population

This review did not impose any restrictions on the population. Individuals of any gender or age were suitable for inclusion.

### Concept

The concept of this review was the translation of physiological data collected by wearable monitoring technologies into clinical decisions such as diagnoses, or early detection related to cardiovascular and respiratory events. Wearable technologies were limited to wrist wearables and intelligent textiles.

### Context

This review considered all publication types, study designs and periods of publication. Studies conducted in either inpatient or outpatient settings were considered for inclusion.

Studies were excluded if they were conducted in a laboratory setting. Studies were also excluded if they focused on a singular component of the technology (*i.e.,* sensor or algorithm design) rather than the use of the technology as an integrated unit. Furthermore, because this review is focused upon cardiovascular and respiratory clinical decisions, studies relating to wearables for clinical decisions in any other domain were also excluded.

## Information sources

To identify potentially relevant literature, a 3-step approach, as previously described by the authors in *Khundaqji et al. (2020b)*, was used. First, a limited preliminary search was conducted in two electronic bibliographic databases relevant to the topic: Ovid Medical Literature Analysis and Retrieval System Online (MEDLINE) and Excerpta Medica database (EMBASE). Following the limited search, an analysis of the titles, abstracts, and indexed terms used was conducted to identify keywords. Subsequently, a second comprehensive search strategy was developed using all identified keywords and index terms by the lead investigator in consultation with a librarian highly experienced in electronic searches. Using the final search strategy, the following bibliographic databases were searched from inception of the database to May 6, 2020: Ovid MEDLINE, EMBASE, Cumulative Index to Nursing and Allied Health Literature, PubMed, Scopus, and SportsDiscus. The search results were exported into EndNote (ver X9.3.3, Clarivate Analytics), with duplicates removed. The Canadian Agency for Drugs and Technologies in Health (CADTH) grey literature searching tool was also used to identify any nonindexed literature of relevance to this review (*Anonymous, 2019*). Finally, the electronic database and grey literature search was supplemented by scanning the reference lists of the included studies.

## Search

The final search strategy for all databases used can be found in Data S1.

## Selection of sources of evidence

Using a priori eligibility criteria, a standardized questionnaire for study selection was developed to assist in the screening of titles, abstracts and full text (Data S2). A pilot exercise preceded each level of screening. Any queries raised by the pilot exercise were reviewed and resulted in the amendment of the questionnaire by the lead investigator. Following the removal of duplicates, the lead investigator screened the papers based on title and abstract. Papers that did not meet the eligibility criteria were removed. Subsequently, the full texts of the remaining papers were retrieved and screened to determine their eligibility. As per the PRISMA guidelines, a flow diagram outlining the study selection process was produced (Fig. 1). A critical appraisal of individual sources of evidence was not undertaken as this scoping review aimed to provide a map of the extent, range, and nature of the existing evidence rather than seek the best available evidence related to practice or policy (*Tricco et al., 2018*).

## Data charting and data items

Firstly, a data-charting form was adapted from the JBI Methodology Guidance for Scoping Reviews at the protocol stage (*Aromataris & Munn, 2020*) (Data S3). Included were key
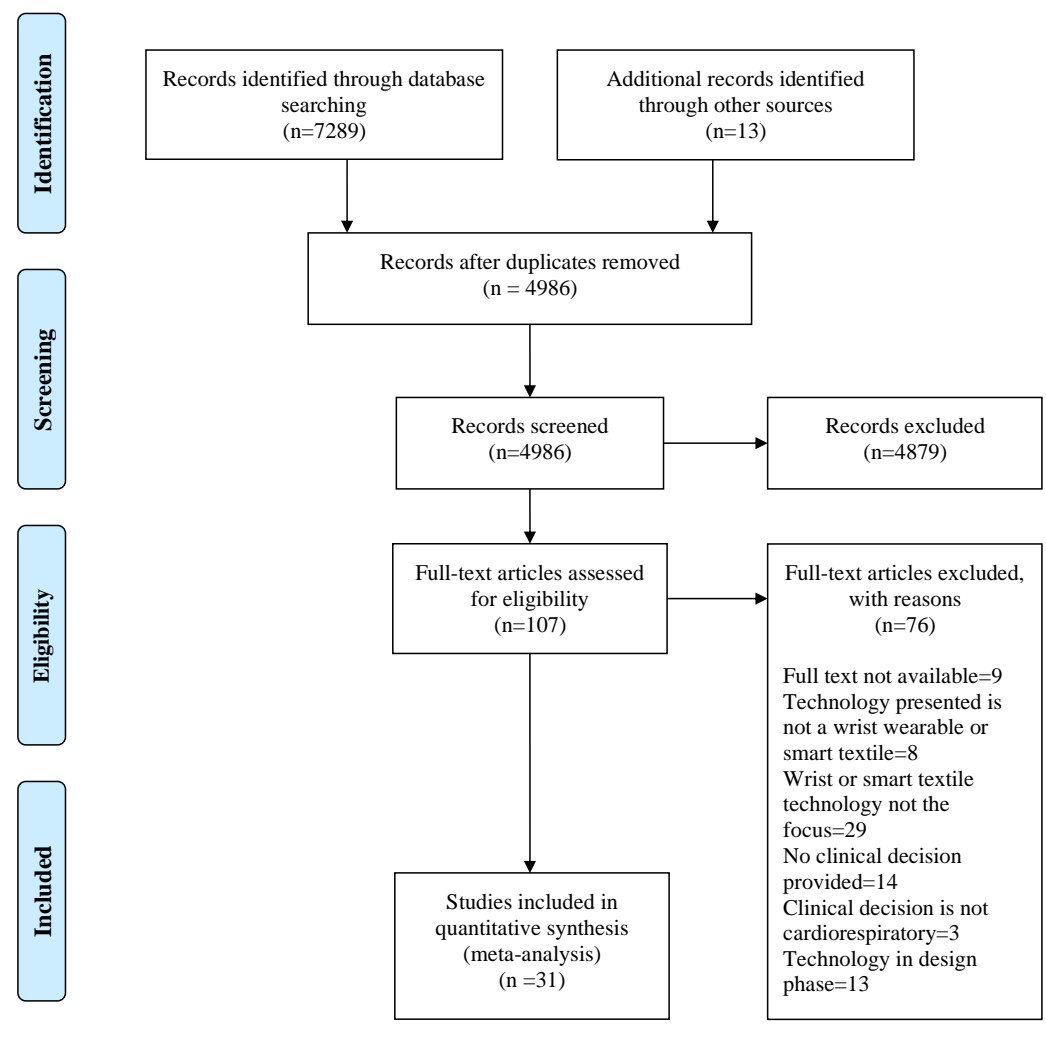

**Figure 1** **Preferred Reporting Items for Systematic Reviews and Meta-Analysis (PRISMA) flow diagram. CINAHL: The Cumulative Index to Nursing and Allied Health; EMBASE: Excerpta Medica database; MEDLINE: Medical Literature Analysis and Retrieval System Online; PubMed.** From *Page et al. (2021)*.

areas of interest, such as study citation details, key study characteristics as well as key findings. Once the form was created, it was tested in a pilot data-charting exercise using 10 studies to ensure that all relevant data were being captured. This exercise resulted in the inclusion of additional fields for the analysis of the statistical performance of the included wearables against a comparator technology (*i.e.,* sensitivity, specificity, accuracy (acc), and positive predictive value (PPV)). Once the testing and refinement of the data-charting form was completed, the lead investigator independently screened all included studies and extracted the relevant information from them. Data extracted in the final data charting form included: study citation details (*i.e.,* author, year of publication, title, reference type, country of origin, and study design), study details (*i.e.,* sample characteristics, study aims, wearable technology used, monitoring conditions, length of monitoring, clinical insights

produced), and key findings (*i.e.,* sensitivity, specificity, acc, and PPV of the wearables performance in clinical decisions compared to a gold-standard technology).

### Synthesis of results

Studies were categorized into the three major study designs identified: (1) observational studies, (2) case reports and series, and (3) reviews. Key study characteristics and findings are graphically represented and tabulated.

## RESULTS

### Selection of sources of evidence

The selection of the sources of evidence followed a methodology previously described by the authors in *Khundaqji et al. (2020b)*. Following the removal of duplicates, a total of 4,986 citations were identified from searches of the electronic databases, the CADTH search tool, and the reference lists of included studies. On the basis of title and abstract screening, 4,879 citations were excluded, of which 107 were retrieved and assessed for eligibility. Of these, 76 were excluded for the following reasons: eight were based on technology that failed the study's definition of a wrist-wearable or intelligent textile, 29 were focused on one aspect of the wearable device (*e.g.,* a sensor, or algorithm) rather than the functional unit, 13 were solely focused on the design of the wearable rather than its application, 14 presented a wearable which did not facilitate nor produce a clinical decision, three presented wearable devices which produced clinical decisions outside of the study's designated cardiorespiratory focus. Additionally, nine full texts were unable to be retrieved by the author.

### General study characteristics

The retrieved papers consisted of two publication types, journal articles (29/31, 93.5%) and conference proceedings (2/31, 6.5%). The year of publication ranged from 2016–2020, with the majority of papers being published in the year 2019 (12/31, 38.7%). Table 1 presents the publication characteristics of the included papers and their associated references. The included papers consisted of six study designs: (1) case reports and series, (2) conference abstracts, (3) editorial comments, (4) experimental cohort study, (5) observational cohort study, and (6) reviews. The countries of origin were made up of 13 countries represented by five continents: (1) Africa (1/31, 3.2%), (2) Asia (5/31, 16.1%), (3) Australia (2/31, 6.5%) (4) Europe (11/31, 35.5%) and (5) North America (12/31, 38.7%). The United States of America produced the most publications out of the 13 countries (12/31, 38.7%). Table 2 presents the countries of origin and study types by the numbers.

### Study designs
#### Observational studies

Observational studies made up 61.3% of the included papers (19/31). Of the 19 papers, nine were studies conducted in the inpatient setting (9/19, 47.4%), nine were conducted in an outpatient setting (9/19, 47.4%) while one study used both an inpatient and outpatient setting (1/19, 5.3%). The most used sample population was made up of patients with atrial fibrillation (AF). Table 3 presents the characteristics of the sample populations

**Table 1 Characteristics of included studies (N = 32).**

| Characteristics | Number of studies, n | Reference(s) |
| --- | --- | --- |
| **Year of publication** | | |
| Before and including 2015 | 0 | N/A |
| 2016 | 3 | *Bonomi et al. (2016), Nemati et al. (2016), Rudner et al. (2016)* |
| 2017 | 4 | *Bonomi et al. (2017), Corino et al. (2017), Hunt & Tanto (2017), Pagola et al. (2017)* |
| 2018 | 6 | *Bonomi et al. (2018), Bumgarner et al. (2018), Eerikäinen et al. (2018), Pagola et al. (2018), Tarniceriu et al. (2018), Tison et al. (2018)* |
| 2019 | 12 | *Dörr et al. (2019), Khatib & Ahmed (2019), Goldstein & Wells (2019), Sajeev, Koshy & Teh (2019), Wasserlauf et al. (2019), Zhang et al. (2019), Avila (2019), Ding et al. (2019), Guo et al. (2019), Hochstadt et al. (2019), Karmen et al. (2019), Perez et al. (2019)* |
| 2020 | 6 | *Chen et al. (2020), Rajakariar et al. (2020), See & Kwong (2020), Walsh & Lin (2020), Ringwald, Crich & Beysard (2020), Yerasi et al. (2020)* |
| **Type of publication** | | |
| Journal article | 29 | *Nemati et al. (2016), Rudner et al. (2016), Corino et al. (2017), Hunt & Tanto (2017), Pagola et al. (2017), Bonomi et al. (2018), Bumgarner et al. (2018), Eerikäinen et al. (2018), Pagola et al. (2018), Tarniceriu et al. (2018), Tison et al. (2018), Dörr et al. (2019), Khatib & Ahmed (2019), Goldstein & Wells (2019), Sajeev, Koshy & Teh (2019), Wasserlauf et al. (2019), Zhang et al. (2019), Avila (2019), Ding et al. (2019), Guo et al. (2019), Hochstadt et al. (2019), Karmen et al. (2019), Perez et al. (2019), Chen et al. (2020), Rajakariar et al. (2020), See & Kwong (2020), Walsh & Lin (2020), Ringwald, Crich & Beysard (2020), Yerasi et al. (2020)* |
| Conference proceeding | 2 | *Bonomi et al. (2016), Bonomi et al. (2017)* |
**Table 2  Countries of origin and study type numbers.**

| Continent | Country (reference) | Total number of studies by country; *n* | Number of studies by type | | | | |
|---|---|---|---|---|---|---|---|
| | | | Case series and report | Conference abstract | Editorial comment | Observational | Review |
| Africa | | | | | | | |
| | South Africa (*Goldstein & Wells (2019)*) | 1 | 1 | | | | |
| Asia | | | | | | | |
| | China (*Chen et al. (2020)*, *Zhang et al. (2019)*, *Guo et al. (2019)*) | 3 | | | | 3 | |
| | Israel (*Hochstadt et al. (2019)*) | 1 | | | | 1 | |
| | United Arab Emirates (*Khatib & Ahmed (2019)*) | 1 | | | | | 1 |
| Australia | | | | | | | |
| | Australia (*Rajakariar et al. (2020)*, *Sajeev, Koshy & Teh (2019)*) | 2 | | | | 1 | 1 |
| Europe | | | | | | | |
| | Germany (*Dörr et al. (2019)*) | 1 | | | | 1 | |
| | Italy (*Corino et al. (2017)*) | 1 | | | | 1 | |
| | Netherlands (*Bonomi et al. (2016)*, *Bonomi et al. (2017)*, *Bonomi et al. (2018)*, *Eerikäinen et al. (2018)*) | 4 | | | | 4 | |
| | Spain (*Pagola et al. (2017)*, *Pagola et al. (2018)*) | 2 | | 1 | | 1 | |
| | Switzerland (*Ringwald, Crich & Beysard (2020)*, *Tarniceriu et al. (2018)*) | 2 | 1 | | | 1 | |
| | United Kingdom (*Hunt & Tanto (2017)*) | 1 | 1 | | | | |
| North America | | | | | | | |
| | United States (*Nemati et al. (2016)*, *See & Kwong (2020)*, *Walsh & Lin (2020)*, *Wasserlauf et al. (2019)*, *Bumgarner et al. (2018)*, *Karmen et al. (2019)*, *Perez et al. (2019)*, *Tison et al. (2018)*, *Ding et al. (2019)*, *Avila (2019)*, *Rudner et al. (2016)*, *Yerasi et al. (2020)*) | 12 | 5 | | 1 | 6 | |
| Total | | 31 | 8 | 1 | 1 | 19 | 2 |

used in the included observational studies. The wristband was the most commonly used wearable (9/19, 47.4%), followed by the smartwatch (SW) (7/19, 36.8%) and the textile wearable Holter (TWH) (1/19, 5.3%). Two studies used both wristbands and smartwatches (2/19, 10.5%). The Apple Watch (Apple Inc., Cupertino, CA) paired with Kardia Band (AliveCor, Mountain View, CA) was the most used wearable (3/19, 5.3%). Among clinical decisions produced by the wearables, AF diagnosis was the most prominent. Other clinical decisions included brady- and tachycardia, other arrhythmias (eg, atrial flutter), as well as normal sinus rhythm. Performance (eg, sensitivity, specificity, accuracy and positive predictive value) was measured against comparators such as the gold-standard 12-lead electrocardiography (ECG) or 12-lead Holter monitor. Table 4 presents the included references, types of wrist and textile wearable devices used, methods of data acquisition, monitoring conditions, clinical event detected as well as their performance against a comparator technology.

### Case control series and reports

Following the observational study design, case series and reports were the most recurrent study design. Seven case reports were identified and one case series which reported on two individual patients. All eight cases reported on patients presenting to the emergency department or clinic with various cardiac complaints. All eight cases reported on the use of wrist wearable devices to assist in clinical diagnosis of a cardiovascular events such as AF, atrial flutter, atrioventricular block, brady- and tachycardia, and ST segment-elevation myocardial infarction (STEMI). The wearable device most reported on was the smartwatch (6/8, 75%), particularly the Apple Watch which was used in 83.3% of studies using a smartwatch. Table 5 presents the key characteristics and findings of the studies and the samples used.

### Reviews

Two narrative reviews were identified and included in the final analysis. The first review aimed to examine the impact of smart wearable devices in early diagnosis, as well as continuous monitoring of cardiovascular disease (Khatib & Ahmed, 2019). The review analyzed the effects of adopting wearable technology on the patient's health and lifestyle as well as the effects of advanced artificial intelligence (AI) in enhancing speed and accuracy of diagnosis. The review also addressed some challenges that smart wearable devices may pose in the future. The second review discussed studies which have reported on the utility and deficiencies of wearable devices in identifying and monitoring cardiac arrhythmias (Sajeev, Koshy & Teh, 2019).

## DISCUSSION

### Principle findings

The primary aim of this review was to systematically analyze the scientific literature concerning the use of wearable devices in the facilitation of clinical decision making in the cardiorespiratory field. The use of wearable devices was restricted to the use of wrist-wearables as well as intelligent textiles. This was primarily due to their larger commercial

**Table 3  Characteristics of the samples used in the included cohort studies.**

| Setting (reference) | Sample population | n | Age (years) | Gender (male %) | Length of monitoring |
|---|---|---|---|---|---|
| Inpatient (*Bumgarner et al., 2018* | Patients with AF presenting for cardioversion procedure. | 100 | 68.2 ± 10.86 | 93% | 30-sec |
| Inpatient (*Chen et al., 2020*) | C1 –NSR (control) C2 - AF | C1-251 C2-150 | C1-59.3 ± 14.8 C2-70.4 ± 11.5 | 52.6% | – |
| Inpatient (*Corino et al., 2017*) | C1 –NSR C2 –AF C3 - ARR | C1-31 C2-30 C3-9 | C1-40 ± 7 C2-76 ± 9 C3-65 ± 15 | 51.4% | 10-min |
| Inpatient (*Ding et al., 2019*) | Patients presenting to cardiology clinic. | 40 | 71 ± 8 | 100% | 42 ± 14-min |
| Inpatient (*Dörr et al., 2019*) | Patients w/pacemaker or ICD | 672 | 76.4 ± 9.5 | 55.7% | 5-min |
| Inpatient (*Hochstadt et al., 2019*) | Patients w/AF | 20 | 74.1 ± 8.7 | 75% | 30-min |
| Inpatient (*Nemati et al., 2016*) | Patients undergoing telemetry. | 46 | 18-89 | N/A | 3.5-8-min |
| Inpatient (*Tarniceriu et al., 2018*) | Post-operative patients. C1 –NSR C2 -AF | C1-15 C2-14 | C1-67.5 ± 10.7 C2-74.8 ± 8.3 | C1-53.3% C2-42.9% | 1.5-h |
| Inpatient (*Rajakariar et al., 2020*) | Patients admitted to medical, cardiac and intensive wards. | 200 | – | – | – |
| Inpatient and Outpatient (*Tison et al., 2018*) | C1 –ECV (inpatient) C2 –Remote (outpatient) | C1-51 C2-1671 | C1-66.1 ± 10.7 C2-N/A | C1–84% C2-N/A | – |
| Outpatient (*Bonomi et al., 2016*) | Patients w/suspected AF/ | 16 | 65.2 ± 14.0 | 63% | 24-h |
| Outpatient (*Bonomi et al., 2017*) | Patients w/cardiac symptoms. | 20 | 67.0 ± 13.0 | 55% | 24-h |
| Outpatient (*Bonomi et al., 2018*) | C1 –AF-patients undergoing ECV. C2 –AF-patients undergoing Holter monitoring. | C1-18 C2-34 | C1-73.1 ± 11.6 C2-67.4 ± 12.1 | C1–56% C2–62% | C1- 1-h pre and post ECV C2- 24-28-h |
| Outpatient (*Eerikäinen et al., 2018*) | C1 –Patients w/AF C2 –NSR | C1-5 C2-10 | C1 - 69 ± 11 C2 - 67 ± 13 | C1-62.5% C2-52.6% | 24-h |
| Outpatient (*Guo et al., 2019*) | Volunteers | 187,912 | 34.7 ± 11.5 | 86.7% | 14-days + |
| Outpatient (*Pagola et al., 2018*) | Patients w/cryptogenic stroke | 146 | 76 | 61% | 28-days |
| Outpatient (*Perez et al., 2019*) | Volunteers | 419,297 | 41 ± 13 | 57% | 113-186 days |
| Outpatient (*Wasserlauf et al., 2019*) | Patients w/suspected AF | 24 | 72.1 ± 7.2 | 65.4% | 110 ± 35.7-days |
| Outpatient (*Zhang et al., 2019*) | Volunteers | C1-263 C2-263 C3-209 | 53.23 ± 13.58 | 50% | 14-days |

Notes.
Abbreviations: AF, atrial fibrillation; ARR, other arrhythmias; C1, cohort 1; C2, cohort 2; C3, cohort 3; ECV, electrical cardioversion; ICD, implantable cardioverter-defibrillator; NSR, normal sinus rhythm.

Khundaqji et al. (2021), *PeerJ*, DOI 10.7717/peerj.12598

**Table 4** Wearable technology used to inform clinical decisions in the included cohort studies and their performance against the gold standard.

| Type of wearable (reference) | Method of data acquisition | Monitoring condition | Clinical event detected | Comparator | Performance | | |
|---|---|---|---|---|---|---|---|
| | | | | | Sensitivity (%) | Specificity (%) | Other |
| Wristband[a] (*Bonomi et al., 2016*) | PPG Accelerometer | Free living | AF | 12-lead ECG Holter | 97 ± 2% | 99 ± 3% | Acc = 98% PPV = 95% |
| Wristband[a] (*Bonomi et al., 2017*) | PPG Accelerometer | Free living | Brady- and tachycardia | 12-lead ECG Holter | Bradycardia –85.0% Tachycardia –89.1% | Bradycardia –99.4% Tachycardia –99.9% | N/A |
| Wristband[a] (*Bonomi et al., 2018*) | PPG Accelerometer | C1 - Supine C2–Free living | AF | C1 –1-lead ECG C2 –12-lead ECG Holter | C1 –97.0% C2 - 93.0% | C1 –100% C2 –100% | Acc ≤ 96% |
| Wristband[b] (*Ding et al., 2019*) | PPG Accelerometer 1-lead ECG | Simulated ADL | Pulse irregularities | 7-lead ECG Holter | 98.2% | 98.1% | Acc = 98.1% |
| Wristband[c] (*Chen et al., 2020*) | PPG 1-lead ECG | Sitting | AF | 12-lead ECG | PPG –88.00% ECG –87.33% Physician review wrist ECG –96.67% | PPG –96.41% ECG –99.20% Physician review wrist ECG –98.01% | Acc PPG = 93.27% Acc ECG = 94.76% Acc Physician review wrist ECG=97.51% |
| Wristband[d] (*Corino et al., 2017*) | PPG Accelerometer | Supine | NSR ARR AF | Not specified. | NSR –77.3% AF –75.4% ARR –75.8% | NSR –92.8% AF –96.3% ARR –76.8% | N/A |
| Wristband[a] (*Eerikäinen et al., 2018*) | PPG Accelerometer | Free living | AF | 12-lead ECG Holter | 98.4% | 98.0% | Acc = 98.1% |
| Wristband[b] (*Nemati et al., 2016*) | PPG Accelerometer 1-lead ECG | Ambulatory | AF | Not specified. | 97% | 94% | Acc=95% |
| Wristband[e] (*Tarniceriu et al., 2018*) | PPG | Supine | AF | ECG | 98.45% | 99.13% | N/A |
| Wristb and[f] Smartwatch 1[g] Smartwatch 2[h] (*Guo et al., 2019*) | PPG | Not specified. | AF | ECG or 24h ECG Holter | 100%[f] 100%[g] 100%[h] | 99.2%[f] 99.2%[g] 98.9%[h] | Acc = 99.2%[f] Acc = 99.2%[g] Acc = 99.1%[h] |
| Wristband[f] Smartwatch 1[g] Smartwatch 2[h] (*Zhang et al., 2019*) | PPG | Free living | AF | 12-lead ECG | 100%[f] 100%[g] 100%[h] | 99.15%[f] 99.16%[g] 98.93%[h] | PPV=93.10%[f] PPV=92.31%[g] PPV=91.67%[h] |
| Smartwatch[i] (*Bumgarner et al., 2018*) | PPG | Seated | AF | 12-lead ECG | 93.0% | 84.0% | N/A |
| Smartwatch[j] (*Dörr et al., 2019*) | PPG | Seated | AF | Mobile ECG | 93.7% | 98.2% | Acc = 96.1% |
| Smartwatch[k] (*Hochstadt et al., 2019*) | PPG | Supine and Seated | AF ARR | ECG | 100% | 93.1% | N/A |

Khundaqji et al. (2021), *PeerJ*, DOI 10.7717/peerj.12598

**Table 4** (*continued*)

| Type of wearable (reference) | Method of data acquisition | Monitoring condition | Clinical event detected | Comparator | Performance | | |
|---|---|---|---|---|---|---|---|
| | | | | | Sensitivity (%) | Specificity (%) | Other |
| Smartwatch[i] (*Wasserlauf et al., 2019*) | PPG | Free living | AF | Insertable Cardiac Monitor (ICM) | 97.5% | N/A | PPV = 39.9% |
| Smartwatch[i] (*Tison et al., 2018*) | PPG | C1 –Sedentary C2 –Ambulatory | AF | C1 –12-lead ECG C2 –Self-reported persistent AF | C1 –98.0% C2 –67.7% | C1 –90.2% C2 –67.6% | C1- PPV = 90.9% C2 –PPV=7.9% |
| Smartwatch[i] (*Perez et al., 2019*) | PPG | Free Living | AF | ECG patch | N/A | N/A | PPV = 84.0% |
| Smartwatch[i] (*Rajakariar et al., 2020*) | 1-lead ECG | Not specified. | NSR Possible AF Unclassifiable | 12-lead ECG | 98.4% | 81.9% | PPV = 98.4% |
| Textile Wearable Holter[m] (*Pagola et al., 2018*) | 3-lead Textile ECG | Free Living | AF | ECG | N/A | N/A | Rate of undiagnosed AF = 21.9% |

**Notes.**

Abbreviations: Acc, accuracy; AF, atrial fibrillation; ARR, other arrhythmias; C1, cohort 1; C2, Cohort 2; N/A, not applicable; NSR, normal sinus rhythm; PPV, positive predictive value; PPG, photoplethysmography.

[a] Philips Cardio and Motion Monitoring Module (CM3 Generation-3, Wearable Sensing Technologies, Philips, Eindhoven, The Netherlands).

Philips Cardio and Motion Monitoring Module (CM3 Generation-3, Connected Sensing; Philips Eindhoven, The Netherlands).

[b] Samsung Simband (Samsung Electronics Co., Ltd., Seoul, South Korea).

[c] Amazfit Health Band 1S (Huami Technology, Anhui, China).

[d] Empatica E4.

[e] PulseOn (PulseOn Oy, Espoo, Finland).

[f] Honor Band 4.

[g] Honor Watch.

[h] Huawei Watch GT (Huawei Technologies Co., Ltd., Shenzen, China).

[i] Apple Watch (Apple Inc., Cupertino, CA) with Kardia Band (AliveCor, Mountain View, CA).

[j] Gear Fit 2 (Samsung Electronics Co., Ltd., Seoul, South Korea).

[k] CardiacSense (CardiacSense, Northern Industrial Park, Caesarea, Isreal).

[l] Apple Watch (Apple Inc., Cupertino, CA) with Cardiogram (Cardiogram Incorporated, San Fransico, CA).

[m] Nuubo Textile Wearable Holter (Nuubo®, Valencia, Spain).

Khundaqji et al. (2021), *PeerJ*, DOI 10.7717/peerj.12598

**Table 5   Summaries of included case reports.**

| Sample | | | Wearable type brand | Data analysis | Clinical decision facilitated | Conclusion |
|---|---|---|---|---|---|---|
| **Subject (reference)** | **n** | **Age (years)** | | | | |
| Male, presenting to ED (*Goldstein & Wells, 2019*) | 1 | 56 | Smartwatch Apple Watch | Retrospective | Atrial Fibrillation | Apple's HR recordings raised suspicion of AF with 2:1 AV block. Diagnosis confirmed with 12-lead ECG. |
| Male, presenting to ED (*Rudner et al., 2016*) | 1 | 42 | Wrist Wearable Fitbit Charge HR | Retrospective | Atrial Flutter | Review of Fitbit data identified the onset of the arrhythmia, permitting ECV and discharge. |
| Female, presenting to ED (*Yerasi et al., 2020*) | 1 | N/A | Wrist Wearable Fitbit | Retrospective | AV Block | Gradual decline of Fitbit HR readings for 2 months prompted ECG investigation leading to 2:1 AV block diagnosis. |
| Male, presenting to clinic (*Karmen et al., 2019*) | 1 | 68 | Smartwatch Apple Watch series 4 | Retrospective | AV Block | Post syncopal episodes, clinic ECG did not provide evidence of AV block. Review of Apple Watch data noted high-grade AV block resulting in urgent biventricular pacemaker implantation. |
| Female, presenting to ED (*Avila, 2019*) | 1 | Middle aged. | Smartwatch Apple Watch | Retrospective | Bradycardia | Retrospective analysis of Apple Watch data helped establish time and length of bradycardia episodes. Patient diagnosed with bradycardia and severely calcified BAV. |
| Males, presenting to ED (*Hunt & Tanto, 2017*) | 2 | Subject 1 –52 Subject 2 –68 | Smartwatch Apple Watch Series 4 | Retrospective | STEMI | Apple's 3-lead ECG matched 12-lead ECG which demonstrated STEMI. Confirmed watch's potential to detect myocardial ischemia. |
| Male, outpatient (*Hunt & Tanto, 2017*) | 1 | 34 | Smartwatch Garmin 630 | Retrospective | Tachycardia | Garmin ECG was capable of capturing tachycardia episode during run. Data interrogated to establish information about timing and length of episode as well as $HR_{max}$ used to aid diagnosis. |
| Male, presenting to ED (*Ringwald, Crich & Beysard, 2020*) | 1 | 45 | Smartwatch Apple Watch | Retrospective | Variant Angina Monomorphic VT | Apple ECG recording during syncopal episode aided in the diagnosis of monomorphic ventricular tachycardia after ED testing and 12-lead ECG were unremarkable. |

**Notes.**

Abbreviations: AF, atrial flutter; AV, atrioventricular; BAV, bicuspid aortic valve; HR, heart rate; $HR_{max}$, maximal heart rate; N/A, not available; STEMI, ST-elevation myocardial infarction; VT, ventricular tachycardia.

availability in comparison to other wearables. The primary outcomes were to (1) provide a clear indication of the types and volume of scientific literature surrounding the use of wearable monitoring devices to produce clinical decisions, (2) summarize the studies completed to date, and (3) identify any knowledge gaps to inform future research. From the 4,986 citations identified after the removal of duplicates, 107 were eligible for full-text review. Of these, 31 were included in the final analysis. Throughout the screening and analysis process, three main study designs were identified: (1) observational studies, (2) case control series and reports, and (3) reviews.

Of the three design types, observational studies were the most prominent, accounting for 61.3% (19/31) of the included papers. All 19 papers used wearable devices for clinical decisions relating to cardiovascular conditions. These conditions included AF, atrial flutter (Aflutter), brady- and tachycardia as well as other arrhythmias, and normal sinus rhythm. No papers using wearable devices to facilitate clinical decisions relating to respiratory conditions were identified. Papers concerning the use of wearable textiles in respiratory populations were identified during preliminary and full-text record screening however, these articles were excluded as they did not meet a key inclusion criteria which required the physiological data collected by the wearables to produce a clinical decision such as the diagnosis or early prediction of a respiratory event. Approximately 50% of the observational studies collected data in an inpatient setting (9/19), while a similar percentage of studies (9/19) collected data in an outpatient setting (*Bumgarner et al., 2018*; *Chen et al., 2020*; *Corino et al., 2017*; *Ding et al., 2019*; *Dörr et al., 2019*; *Hochstadt et al., 2019*; *Nemati et al., 2016*; *Tarniceriu et al., 2018*; *Rajakariar et al., 2020*; *Bonomi et al., 2016*; *Bonomi et al., 2017*; *Bonomi et al., 2018*; *Eerikäinen et al., 2018*; *Guo et al., 2019*; *Pagola et al., 2018*; *Perez et al., 2019*; *Wasserlauf et al., 2019*; *Zhang et al., 2019*). One study collected data in both an inpatient and outpatient setting (1/19) (*Tison et al., 2018*). Individuals with suspected or diagnosed AF made up the majority of cohorts. One limitation identified across the included papers was the absence of the key demographic characteristics of the samples used such as skin tone and variables such as body mass index (BMI). These variables have been reported to influence PPG measurements by introducing noise, therefore effecting generalizability of the findings (*Nelson et al., 2020*). This is particularly important as the wearable type most used to monitor samples and facilitate or produce clinical decisions was the wristband, followed by the SW and the TWH. Both the wristband and SW used similar data acquisition techniques with PPG being the standard method of measuring pulse rate. In addition to PPG, some wristbands and SW's also employed a 3-axis accelerometer, 1-lead ECG or both. The wrist acceleration signal was used by the wearables algorithm to calculate the amount of motion associated with each inter-pulse interval (IPI). In case the motion intensity associated with a certain IPI exceeded a predefined threshold, that IPI was labelled invalid and discarded from the analysis (*Bonomi et al., 2018*). The use of the 3-axis accelerometers is particularly important when monitoring ambulatory individuals where displacement of the PPG sensor over the skin, changes in skin deformation, blood flow dynamics, and ambient temperature may cause motion artifacts (*Bent et al., 2020*). The included studies used both ambulatory (9/19) and sedentary samples (6/19). Two studies used both ambulatory and seated participants while 2 more did not specify the monitoring
condition (*Bonomi et al., 2018*; *Tison et al., 2018*). Under ambulatory conditions, the wearable device's performance in producing clinical decisions was compared against a 12-lead or 7-lead ECG Holter monitor. Under sedentary conditions the wearable device's performance was compared against the gold standard 12-lead ECG or an insertable cardiac monitor (ICM). Another study compared the use of wearables for the diagnosis of AF using both ambulatory and static samples and found that sensitivity, specificity, accuracy and PPV were greatly improved in the latter (*Tison et al., 2018*). In the sedentary sample a sensitivity of 98%, specificity of 90.2% and a PPV of 90.9% were reported compared to 67.7%, 67.6% and 7.9% in the ambulatory sample.

The second study design identified was the case control series and reports. These studies were made up seven case reports, and one case series. The seven case reports each described an individual case where a wrist monitoring device was used to facilitate or support the clinical diagnosis of a cardiovascular event such as AF, Aflutter, AV block, bradycardia and ventricular tachycardia (*Goldstein & Wells, 2019*; *Rudner et al., 2016*; *Walsh & Lin, 2020*; *Yerasi et al., 2020*; *Karmen et al., 2019*; *Hunt & Tanto, 2017*; *Ringwald, Crich & Beysard, 2020*). Similarly, the one case series reported on the use of wrist-wearables to support the diagnosis of a STEMI in two separate cases (*Avila, 2019*). In all cases, the diagnosis was made by a specialist physician after retrospective analysis of the wearable device's raw data. Despite the current need for physician analysis of the device's data for a diagnosis, the ability of wearables to collect the necessary data accurately highlights their future potential in real-time diagnosis to prompt further specialist investigation. The wearable used most was the Apple Watch, particularly the Series 4 which is equipped with both PPG and a single lead (lead I) ECG. Other wearables included the Fitbit Charge HR and the Garmin 630 Running Watch. It is of note that seven of the eight cases reported on male patients (*Goldstein & Wells, 2019*; *Rudner et al., 2016*; *Yerasi et al., 2020*; *Karmen et al., 2019*; *Hunt & Tanto, 2017*; *Ringwald, Crich & Beysard, 2020*; *Avila, 2019*). This is important as research has reported higher device errors in males compared to females when measuring HR and energy expenditure (EE) in commercially available wrist-worn devices (*Nelson et al., 2020*; *Shcherbina et al., 2017*).

Lastly, two narrative reviews were identified in this study. The first review aimed to examine the impact of smart wearable devices in early diagnosis, as well as continuous monitoring of cardiovascular disease (*Khatib & Ahmed, 2019*). The second review aimed to examine and report on the utility and deficiencies of wearable devices in identifying and monitoring cardiac arrhythmias (*Sajeev, Koshy & Teh, 2019*). Neither review reported on the statistical performance of wearable devices against a comparator in facilitating clinical decisions.

## Gaps in the literature

As indicated in the principle findings a large disparity exists between the use of wrist wearables and intelligent textiles for the facilitation of cardiovascular and respiratory clinical decisions. The lack of use of wrist and textile wearables to facilitate respiratory clinical decisions is likely due to the current limitations of the available sensors as well as their limited commercial availability. Currently, direct and unobtrusive respiratory

measures exist through intelligent textiles by integrating sensors such as respiratory inductance plethysmography (RIP), piezoresistive and bio-impedance into the textile in the form of bands or electrodes (*Khundaqji et al., 2020b*; *Curone et al., 2010*; *Di Rienzo et al., 2011*; *Di Rienzo et al., 2005*; *Magenes et al., 2009*; *Paradiso & De Rossi, 2008*; *Paradiso, Loriga & Taccini, 2005*; *Steinberg et al., 2019*; *Zhang et al., 2009*). RIP sensors measures ventilation by assessing thoraco-abdominal pressure changes using transducer recording bands (*Marlin & Deaton, 2007*). This is accomplished by placing the bands at the level of the nipples and at the umbilicus to monitor cross-sectional changes reflected by changes in inductance or resistance to change in flow of the transducers (*Mehra & Strohl, 2009*). Similarly, piezoresistive sensors, often in the form of bands, detect respiratory measures like RR through their positioning around the thoraco-abdominal compartments to detect changes in the cross-sectional area (*Paradiso, Loriga & Taccini, 2005*). Bio-impedance sensors provide an alternative method to the monitoring of respiratory activity based on the changing impedance of the thorax using superficial electrodes (*Pacela, 1966*). Although studies have shown intelligent textiles to be largely valid in determining respiratory parameters such as RR at rest and sub-maximal intensities, they have demonstrated variable validity when measuring other parameters such as minute ventilation ($V_E$) and tidal volume ($V_T$) (*Clarenbach et al., 2005*; *Elliot, Hamlin & Lizamore, 2019*; *Smith et al., 2019*). A significant limitation of these sensors is motion artefacts produced by movement. To ensure optimal signal acquisition through continuous conductivity, sensors must maintain constant and unimpeded contact with the skin. Contact between the sensors and the skin may often be disrupted by excessive body movement which, in turn, may disrupt signal acquisition through the introduction of noise (*Elliot, Hamlin & Lizamore, 2019*; *Smith et al., 2019*; *Montoye, Mitrzyk & Molesky, 2017*). This issue has likely limited the widescale adoption of respiratory sensors outside controlled settings and therefore limited the study of their use in field research to produce crucial clinical decisions. To date, the majority of studies concerning respiratory sensors in intelligent textiles continue to be on prototype design and validation (*Khundaqji et al., 2020b*). While the study of photoplethysmography (PPG) to monitor RR exist, studies analyzing the validity of PPG in wrist wearables to measure RR are limited. This is likely a result of PPG signals acquired at the wrist being of lesser quality than those acquired in positions like the chest as the optical sensors are sensitive to wrist movements (*Jarchi et al., 2018*). This lack of investigation of wrist PPG to monitor RR may explain why limited studies currently exist using wrist wearables to provide respiratory field data which is translatable into clinical decisions. However, with the monitoring of parameters such as RR being important indicators of severe respiratory disease, highlighted even more so by the COVID-19 pandemic, emerging research in design and validity, particularly in intelligent textiles, point to its eventual readiness for commercial adoption and use in the clinical domain .

Furthermore, the principle findings point to the need for further research into wearables capable of clinical decision making in cardiovascular conditions other than AF. To date, AF has been the most investigated condition with wrist wearables.

Lastly, there seems to be a lack of standardization in the research when reporting on samples used. As discussed earlier, many variables such as skin tone, BMI, and body hair

density are capable of introducing noise into the sensors and algorithms. By not reporting these characteristics there is a loss of a generalizability of the results. This provides the opportunity for the development of a standardized reporting method. Additionally, the majority of study samples were made up of primarily males. Previous research into wrist wearables has reported on their varying performance when used by males and females, indicating there remains a need to study these wearables in a female population to better understand their capabilities and limitations (*Shcherbina et al., 2017*).

## Limitations

This scoping review was limited to publications in English, which may have excluded key papers published in other languages. Although larger studies such as those included from China are typically published in English there may have been others published in different languages. This review was also limited to clinical decisions produced or facilitated by wrist wearables and intelligent textiles in the cardiovascular and respiratory domains. This was due to wrist wearables and intelligent textiles being among the most commercially available and cardiovascular and respiratory conditions the most reported on. Lastly, the screening, inclusion, exclusion and data charting stages of this review were conducted by one investigator (HK), which could have reduced the likelihood that all relevant papers were identified. This could have also resulted in reviewer bias. However, the chances of this were reduced by performing pre-screening testing and exercises to reduce any bias.

## Comparisons with previous work

To the authors' knowledge, this review was the first to systematically map the scientific literature pertaining to the use of wrist wearable and intelligent textiles in the making or facilitation of clinical decisions across both the cardiovascular and respiratory domains. Only two other narrative reviews, which were included in the results, investigated the use of wrist wearables for identifying and monitoring cardiac arrhythmias. However as previously discussed these reviews did not report on the performance of the wearables in making or facilitating clinical decisions against a gold standard.

## CONCLUSIONS

The transition towards personalized, and evidence-based healthcare has prompted the rapid advancements in wearable devices and mobile cloud computing technologies to off-load burden from healthcare systems, reduce costs and improve health outcomes. However, to date, the bulk of commercially available wearable wrist and textile technology is targeted towards general health and fitness. This scoping review systematically surveyed the existing body of scientific literature surrounding the use of wrist and textile monitoring technology to facilitate clinical decisions in the cardiovascular and respiratory domains. Through this review it was identified that to date, all studies employing wrist and textile technology in the clinical setting are focused on the diagnosis or early prediction of cardiovascular events such as AF and other arrhythmias in male dominant samples. Currently, the majority of these technologies require the retrospective analysis of the wearable's raw data to confirm diagnosis by a physician. These technologies demonstrated variable statistical performance

in comparison to gold-standard depending on type of technology, sample characteristics and setting in which they were employed. When performance was tested in sedentary and ambulatory cohorts, sensitivity and specificity were superior in the former. Research is still needed in the respiratory domain, female dominant cohorts and more diverse samples that better report on variables such as skin tone, hair density and BMI.

### Funding
The authors received no funding for this work.

### Competing Interests
Mike Climstein is an Academic Editor for PeerJ.

### Author Contributions
- Hamzeh Khundaqji conceived and designed the experiments, performed the experiments, analyzed the data, prepared figures and/or tables, authored or reviewed drafts of the paper, and approved the final draft.
- Wayne Hing, James Furness and Mike Climstein conceived and designed the experiments, authored or reviewed drafts of the paper, and approved the final draft.

### Data Availability
The search strategies used for each electronic data base searched are available in the Supplementary File.

### Supplemental Information
Supplemental information for this article can be found online at http://dx.doi.org/10.7717/peerj.12598#supplemental-information.

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
