# Peer review of "Wearable technology to inform the prediction and diagnosis of cardiorespiratory events: a scoping review"

_PeerJ, doi:10.7717/peerj.12598_

## Round 0.1 · original submission · Major Revisions

This is a well-written manuscript and addresses an important and growing area in healthcare delivery. There are several issues that require addressing. The most pressing issue, as outlined by Reviewer 2, is to ensure that relevant studies related to wearables and respiratory conditions have not been missed. There is very little reference and discussion throughout the manuscript concerning respiratory monitoring - this needs to be addressed. Please ensure that your search has been comprehensively undertaken and is broad enough to detect relevant respiratory studies. Further discussion should be added concerning the role of wearable technology and respiratory conditions.

Reviewer 1 ·

Basic reporting

In this article, Khungaqji et al provide a systematic review of how physiological data from wearable technology is used to assist clinical decision making. The aim of the review was to map out the current body of literature to identify knowledge gaps and thereby inform future research in the cardio-respiratory field. The English used is proficient, the references appear appropriate and the rationale for the study is presented clearly.

Experimental design

The methods used for the literature review appear appropriately described and executed. The review appears appropriately organised.

Validity of the findings

Within the introduction, the authors correctly state that the literature to date has focussed on the design, reliability and validity of these devices in controlled settings, and that the next phase is to translate this data into clinical useful decision making.
The conclusion states that although wearables are able to accurately collect physiological data, there is a need for specialist physicians to review the raw data before making a definitive diagnosis. They also comment on the lack of data from the respiratory domain, and a large bias towards AF detection in the cardiovascular domain. These are all correct, and point towards the current limitations in both sensor technology and the software algorithms designed to interpret the raw data which they provide.

Additional comments

The article is clearly written and the methodology, findings and conclusions are all appropriate. The discussion provides further details about the current gaps in the literature and also some of the limitations of the current technologies. In terms of answering the aims of the review, in providing a detailed map of the current literature, the authors have successfully identified current knowledge gaps and areas for future research including: 1) Application of wearable devices for clinical decision making; 2) Extending the use of these devices outside the diagnosis of AF within the cardiovascular domain and 3) The use of wearable devices for respiratory conditions. The latter appears to be an area with a significant knowledge gap.

·

Basic reporting

No comment.

Experimental design

1. The authors are addressing a timely issue, in wearable tech and cardiorespiratory decision making. On the face of it, this number of articles seems quite low for my anecdotal observations of the literature, as well as clinical practice. For example, our study about smart watch and afib was recently published but not included here (10.1001/jamanetworkopen.2021.5821). I am not requesting you cite my work (it may fall out of the study period), it simply provides an immediate example of my concerns related to the search. I provide additional issues related to respiratory below. Also, health systems are suing "hospital at home" nowadays- would these types of articles provide relevant articles?

2. Study reports no respiratory studies, which seems incorrect. On quick Google search, I found doi: 10.2196/10046. Is this study applicable? Search criteria includes "respiratory" but not "pulmonary," which could be an issue. Also see DOI: 10.2147/COPD.S193037. Also, COVID-specific searches could change the number of resulting manuscripts.

3. Clinical decision making is part of the search. I think this needs to be better defined in the text. For example, decision making by the clinician? Decision making by the patient or someone lese? I wonder if this part of the query really limits the results.

Validity of the findings

4. "To date, studies employing wearables to facilitate clinical decisions have largely focused
upon the cardiovascular domain." This does not seem like a conclusion since the denominator is limited to CV studies.

Additional comments

5. The biggest gap is prospective, randomized studies link to health outcomes, which was not mentioned in "gaps." Also, what about issues surrounding integration with electronic health records/interoperability?

---

## Round 0.2 · Minor Revisions

Thank you for addressing the reviewer comments. I have only one minor revision required. In Table 4, for references 4 and 13, it is unclear what parameter the 'other' column is referring to. Please address this. Otherwise, I believe the manuscript is well written has addressed its aims in undertaking a scoping review of this subject.

---

## Round 0.3 · accepted · Accept

Thank you for addressing the suggested revision.